# Effect of the Technological Process from Vine to Wine on Pesticide Residues in Vernaccia di Oristano Cultivar

**DOI:** 10.3390/foods10061295

**Published:** 2021-06-04

**Authors:** Francesco Corrias, Riccardo Taddeo, Nicola Arru, Alberto Angioni

**Affiliations:** Food Toxicology Unit, Department of Life and Environmental Science, University Campus of Monserrato, University of Cagliari, SS 554, 09042 Cagliari, Italy; francesco.corrias@unica.it (F.C.); ricky.taddeo@hotmail.it (R.T.); nicola.arru.logica@gmail.com (N.A.)

**Keywords:** vinification, grapes, must, wine, Liquid Chromatography with tandem mass spectrometry, pesticide residues

## Abstract

Vernaccia is a white grape mostly used to produce a distinct wine protected by the controlled designation of origin (DOC) recognition. It is very susceptible to fungal disease, and it is subjected to a defined management protocol in the field. Winemaking could influence pesticide residues through different mechanisms. This work investigated the influence on pesticide residues of the winemaking process at the industrial level of the wine Vernaccia di Oristano. Thirty-five samples of grapes, two musts, and two vines (both liquid fraction and pellets) were analyzed by using a validated multiresidue LC-MS/MS (Liquid Chromatography with tandem mass spectrometry) method. Data obtained showed the presence in grapes juice of 16 pesticides (8 not allowed in the EU) with mandipropamid and mepanipyrim, the only ones with values higher than their MRL (maximum residue level). Pesticide residues decrease in must was related to the dilution effect due to mixing the grape samples. However, pellets analysis also confirmed the high affinity of pesticides for the suspended material (fenhexamid), whereas the increase in wine to a re-solubilization process from the lees during the fermentation step. The present paper highlighted the effectiveness of the technological process of winemaking to decrease pesticide residues compared to the raw material.

## 1. Introduction

Vernaccia di Oristano is a Sardinian (Italy) white grape cultivar used to produces different wines, which follow the tastes and aromas of Sherry, where almond blossoms with bitter tastes are present [1]. It is a very ancient cultivar, reported before 1327 in the Breve Villa di Chiesa [2] and the Carta del Logu [3] and formally protected by the municipality of Iglesias. Despite the restricted cultivation area, which involves 17 municipalities in the Oristano district, it has a leading role in the Sardinian wine scene because of its unique flavor characteristics, which allowed it to be recognized with the denomination of controlled origin (DOC) [4].

The harvest is usually carried out from the second half of September until the first 10 days of October, and grapes must have a sugar content of at least 21–22° Babo. The vinification of Vernaccia is a typical white wine vinification with soft pressing; the obtained juice is sent to the barrel for fermentation. The decanting step (at least two) and the necessary clarifications are then carried out. During March following the harvest, the wine is transferred to chestnut or oak barrels with thin slats, left full for about 20% of their capacity. The biological refinement process follows this by the film-forming yeasts (flor); refinement continues with ageing in the oxidative phase. The major adversities of the cultivation of Vernaccia grapes are represented by the fungi *Uncinula necator* and *Plasmopara viticola*, which are responsible for the white disease and grape blight, respectively [5]. These diseases affect all the plant’s green organs, causing significant damage to the leaves, the bunches, and the single berries. They are two of the most critical cryptogamic diseases of vines, spread throughout Europe.

Moreover, fly attack by *Lobesia botrana, Planococcus ficus,* and *Tropinota squalida* [5] can be detected on Vernaccia fields. Plant protection products (PPP) are used in early-season disease infections, following integrated pest management (IPM) strategies [6]. However, the high susceptibility of Vernaccia grapes can lead farmers to carry out more treatments regardless of the indications of the IPM, and multiresidue pesticide analysis is needed to ensure food safety; when the product does not comply with the legal limit, it is discarded. The technological process can influence the amount of pesticide residues in the final product. When vinification is carried out, two fermentative processes are involved. The former transform sugars into alcohol, whereas the second malic into lactic acid. Alcoholic fermentation is yeast mediated, whereas bacteria run malo-lactic fermentation. Microorganism proliferation could be affected by pesticide residues and vice versa. Many studies showed that yeasts and bacteria could decrease pesticide residues by degradation and adsorption [7,8,9,10]. Moreover, lees and cake showed an affinity for many pesticides reducing the residues in the liquid phase and enhancing food health [11,12].

This paper reports a comprehensive study on pesticide residues contamination before and after the industrial processing of grapes collected in the field to produce Vernaccia wine. The analytical determination of pesticide residues was carried out using a validated UHPLC-MS/MS multiresidues method integrated with a modified QuEChERS extraction step [13].

## 2. Materials and Methods

### 2.1. Samples Collection and Processing

Trials were carried out in 300 ha located in the province of Oristano in Sardinia in August 2019. Thirty-five fields were selected from those that usually supplied grapes to the Industry. Raw grapes samples were harvested from each selected field with a potential alcohol content >15% and represented all field production. After that, samples were brought to the winery on two different days and processed at the industrial level. Grapes from fields 1 to 23 were merged in the fermentation vat V30, whereas grapes 24–35 were merged in the vat V31; each vat provided one wine W30 and W31. Briefly, the grapes were destemmed via a mechanical device that works like a cylindrical coin counter that rotates on its side; holes were big enough for grapes to fall through for collection, whereas the stems remain inside and are collected separately. Horizontal bladder presses allow a soft and gently pressing of grapes which keeps the aroma intact. Grape berries are loaded into the bladder, which inflates and presses against the wall of the tube. Over time, the pressure increases, and the juice is separated from skins and seeds and set aside. Fermentation is carried at 20 °C to protect fruit aromas. Ten random aliquots of 1 kg from each harvest were merged, destemmed and homogenized. After that, three grape juice samples (1L) were collected and used for pesticide analysis. Thirty-five samples of grape juice, two merged musts, and two wines were collected and transported to the laboratory. Field treatments followed the integrated production strategy set in Sardinia for Vernaccia cultivar. At the end of white wine fermentation, the temperature is dropped to a respectably chilly level, and the dead yeasts settle to the bottom of the tank. Specialized technicians supervised plant protection plans to use, among the authorized pesticides, those with the shortest pre-harvest interval, lowest toxicity, and minimum environmental persistence [14].

### 2.2. Chemicals and Reagents

Analytical standards (≥99.5% purity) were bought from Dr Ehrenstorfer (Lab service Analitica, Milan, Italy) (Table 1). Acetonitrile (ACN) and methanol (MeOH) were LC/MS grade solvents (Sigma Aldrich, Milan, Italy). CH_2_O_2_ was reagent grade (>95%, Honeywell, Sigma Aldrich), NH_4_HCO_2_ solution 5 M (0.315 g mL^−1^) (G1946-85021, Agilent Technologies). 4 g MgSO_4_, 4.1 g NaCl, 1 g C_6_H_5_Na_3_O_7_ * 2 H_2_O, 0.5 g C_6_H_8_Na_2_O_8_ (No: 5982—6650, En Method 15662, Agilent Technologies, Milan, Italy) and 150 mg PSA, 900 mg MgSO_4_ (No: 5982–5056, EN Method, fruit and vegetable, Agilent Technologies, Milan, Italy) were used for the QuEChERS extraction step [15].

MilliQ water from a Millipore purification system (MilliQ integral, Merck, Milan, Italy) was used with a conductivity below 18.2 MΩ. Stock standard solutions of pesticides were prepared at ~1000 mg L^−1^) in ACN. Working solutions were prepared daily by diluting the stock solutions with the eluent mixture (e.m.) at T = 0.

### 2.3. Sample Preparation

Samples of grape juice, must, and wine were collected directly from the industrial processing plant, brought to the laboratory and subjected to the analytical processing step. Twenty mL of each homogenized sample were put in a 50 mL test tube and centrifuged for 15 min at 3154× *g* and 10 °C (Centrifuge 5810 R, Eppendorf AG 22,331 Hamburg). The liquid phase was then separated from the pellet. 5 mL of sample and processed according to Corrias et al. (2020) [13]. The pellet was directly extracted with 10 mL of ACN agitated in a vortex for 1 min and in a rotatory shaker for 15 min; after centrifugation, 6 mL of the supernatant were recovered and transferred to a 15 mL test tube having 1 g of QuEChERS salts (part no. 5982–5056, Agilent, Milan, Italy). The organic solutions were filtered at 0.45 µm (PTFE, Thermo Scientific) and transferred in a 1.8 mL vial for LC-MS/MS analysis. UHPLC-MS/MS analysis were carried out according to Corrias et al. (2020 [13] (Table 1)).

### 2.4. Method Validation

An earlier validated analytical method was adapted to grapes, must and wine [13]. For recovery assay, organic white grapes and wine sample were bought from a local organic farm; the grapes were destemmed and crushed in the laboratory to obtain the resulting must. The grape juice, must, and wine samples were spiked at LOQ and 5 × LOQ levels (three replicates *n* = 18). Six blank control samples for each investigated matrix were spiked with a mixed multiresidue standard at 3 × LOQ and analyzed in one day for intraday repeatability (RSDr, *n* = 18), and two samples in six separate days (*n* = 36) were used for reproducibility (RSD_wR_). Each sample belonged to an independent experiment. The instrumental sequence was conducted according to SANTE indications [16]. The analytical response of the active ingredients in the eluent mixture (e.m.) and matrix extract was used to define the matrix effect. The absence of MRM peaks at the retention time of the a.i. was used to assess method selectivity. Linearity was admitted as acceptable when the coefficient of determination (r^2^) was above 0.990. The expanded measurement uncertainty (U), was calculated with a level of confidence of 95% using the equations
u′ = √u′(bias)^2^ + u′(precision)^2^; (1)
U= k × u′(2)

The instrumental LOD and LOQ were calculated as three and ten times the signal to noise ratio (S/N) [17].

## 3. Results and Discussion

### 3.1. Validation Method

The UHPLC-MS/MS method allowed to investigate 115 pesticide residues in grape juice, must and wine, of which 64 authorized in wine viticulture [18].

Validation data are the mean of the single values obtained for grape juice, must and wine (Table 1). RSD% values were below 20% for each pesticide, showing minor variations among the three matrices. Calibration curves prepared in pure solvents and blank matrix (grape juice, must, and wine) showed superimposable values with correlation coefficient (r^2^) ranging from 0.9955 to 1.0000 and RSD% max 2.08%. Linearity was above the condition set for method validation (Table 1). The QuEChERS method allowed an acceptable purification of the extracts, and no interfering peaks were detected in the chromatographic range of interest (Figure 1).

Complying with SANTE principles, apparent recovery (this term is recommended when the observed value is obtained via an analytical procedure such as a calibration graph) [19] data ranged from 75.6% to 115.9% at LOQ level and from 75.9% to 110.7% at 5 × LOQ. The minimum and maximum coefficient of variability ranged from 1.4% to 17.7% (Table 1). Repeatability and within laboratory reproducibility showed satisfactory results below 18%, with maximum and minimum RSD% of 17.7% and 5.4% in RSDr and 18.0% and 4.8% in RSDwR (Table 1). The expanded uncertainty (U) calculated from average recoveries and RSD_wR_, was for all pesticides below 50% of the default values. The LC-MS/MS method showed good robustness and could be used to analyze the pesticides in grapes juice, must, and wine (Table 2 and Table 3). The instrument limits of quantification (LOQs) and of determination (LODs) were below the MRLs set by the European Community for wine grapes [20,21] (Table 1).

### 3.2. Sample Analysis

Thirty-five samples were collected from the fields representing 39,400 kg; the grape yield was 76%. Therefore, assuming that all pesticide residues in the grapes were transferred in the must, it should be reasonable to increase the concentration of pesticide residues by a factor of 1.3 in the wine.

The analysis of the grape juice from fields 1–23 allowed the identification of 16 pesticides among the 115 searched with the above MRM method (Figure 1).

The distribution among samples was uneven, with a frequency (F) of pesticide ranging from 1 to 23 in the worst case. Mandipropamid (23F) and fenhexamid (22F) were the most represented, followed by azinphos ethyl and mepanipyrim (16F), acephate, formetanate, pymetrozine, and carbendazim (13F), zoxamide, emamectine benzoate, and tau-fluvalinate (10F) (Table 2).

In all cases, the values found in grapes were below the limit of quantification (LOQ) except for mandipropamid, fenhexamid, mepanipyrim, iprodione, zoxamide, spinosad, and etofenprox. The pesticides accounting for a total residue >LOQ, showed values significantly lower than the MRL set for wine grapes, except for mandipropamid (3F, MRL 2.00 mg kg^−1^) and mepanipyrim (1F, MRL 2.00 mg kg^−1^), which showed values from 2.99 to 6.60 mg kg^−1^, and of 2.36 ± 1.15 mg kg^−1^, respectively.

The samples 24–35 showed similar behavior with 15 pesticides above the LOD of the method. Although, compared to the samples from the previous harvest, they showed a lower frequency of distribution. Mandipropamid and fenhexamid (6F) were the most represented, followed by zoxamide, spinosad, emamectine benzoate, and tau-fluvalinate (5F), any residues of iprovalicarb and etofenprox was detected in this batch. Only eight pesticides showed residue values above the LOQ of the method but consistently below the MRL set for wine grapes. Mandipropamid showed three samples with values above the MRL (4.88–10.56 mg kg^−1^) (Table 3).

Pesticide residues values in the musts (V30 and V31) were lower than those found in the single grapes juice samples and below the instrumental LOQ. Mandipropamid (1.05 ± 0.61 mg kg^−1^ ± RSD%) and zoxamide (0.02 ± 2.06 mg kg^−1^ ± RSD%) in V30 (Table 2), and mandipropamid (0.75 ± 3.99 mg kg^−1^ ± RSD%) zoxamide and emamectin benzoate (max 0.02 mg kg^−1^) in V31, were the only pesticide quantifiable (Table 3). The decrease in must juice is related to the dilution effect due to mixing the grape samples. The corresponding wines W30 and W31 showed for these three pesticides values comparable to the must.

The grape juices, musts, and wines were subjected to centrifugation before pesticide analysis to separate the suspended material. The obtained pellets accounted for 0.25 ± 18.79 mg L^−1^ ± RSD% in the grapes, 0.34 ± 8.32 in the must and 0.02 ± 0.10 mg L^−1^ ± RSD% in the wine. The residues detected in the pellet showed high variability among the different matrices. Mandipropamid was below LOQ in the grapes pellet while showed increasing values passing from must to wine; a similar trend was detected for fenhexamid, zoxamide, spinosad, emamectin benzoate, and tau-fluvalinate in both wines and iprodione in V30-W30 (Table 2 and Table 3).

Pesticide residues found in the pellet agreed with previous studies reporting the adsorbing effect of the solid fraction during winemaking on pesticides [12]. Moreover, the increasing values found in the wine pellet can be explained by considering the conversion from an aqueous-sugar solution (must) into a hydroalcoholic solution (wine) during fermentation with a re-solubilization of pesticides from the lees.

The persistence of pesticides applied in the field is related to the intrinsic chemical characteristic of the compounds as well as on environmental conditions such as sunlight exposure (photodegradation), temperature (thermodegradation), plant activity (enzymatic degradation), and co-distillation during plant physiological respiration [22,23]. Food processing techniques may increase or decrease pesticide residues in the final products [13,24,25]. During the fermentation step, the pesticide residues on grapes can be transferred to the must, influencing the selection and development of yeast strains. In reverse, yeasts can actively reduce the levels of the pesticides or adsorb them on the lees. Several papers reported the behavior from vine to wine of selected pesticides and the influence of yeast on their residues [7,8,9,10,11,12,13,26,27,28,29,30]. Xu et al. [31] reported that mandipropamid residues were higher in the pomace than the juice during winemaking. Likewise, Cus et al. [29] reported that fenhexamid remained in the lees, and no residue was found in the liquid phase. Đorđević and Durovic-Pejcev [25] highlighted the affinity of pesticide for skin and pulp when fruits are subjected to juicing during the pressing step of grapes. Moreover, during winemaking, there is a significant pesticide reduction due to the adsorbing effect of the lees [12].

According to EFSA (2018) [32], the pesticide processing factor was calculated for the pesticide quantified in grape juice vs. wine (PF1) and must vs. wine (PF2).

All pesticide showed a PF1 < 1 indicating a decrease of the residues during the processing, whereas they showed a PF2 > 1 confirming the hydroalcoholic media ability to extract pesticide from the lees (Table 2 and Table 3).

Among the 16 pesticides found in grape juice samples, must, and wine, eight were not allowed in wine grapes in the EU (Table 2 and Table 3). However, their residues were far below the MRL set in fruit samples, and most cases, below the LOQ of the method. These pesticides could be attributed to pollution from treatments in the neighboring fields (run-off) or polluted water supply [33,34].

## 4. Conclusions

This paper reported the behavior of pesticide residues from field management of Vernaccia vineyards to produce white wine at the industrial level. The LC-MS/MS method was adapted to analyze 115 pesticide residues in grapes, must, and wine. The LOQ and LOD were suitable to quantify pesticide residues also in trace in the different matrices. Grapes showed 16 pesticides with values below the MRL; however, some samples showed values of mandipropamid 6 times higher than the MRL set for grapes and one time for mepanipyrim. The corresponding must showed a significant decrease of the residue, related to a dilution effect, while the wine showed a similar residue of the must, indicating no adsorbing effect by the lees. On the contrary, Fenhexamid confirmed its attitude to be adsorbed by the lees and the suspended particulate. Moreover, in the grapes, we also found eight active ingredients not allowed in the EU, below the LOD in the wine.

The present paper highlighted the effectiveness of the technological process of winemaking to decrease pesticide residues compared to the raw material. Moreover, even if vineyards are not conduced in good agriculture practice (GAP), the technological process could reduce pesticide residues. Some pesticide can be resolubilized in the hydroalcoholic solution from the particulate, and their use should be avoided before harvest.

## Figures and Tables

**Figure 1 foods-10-01295-f001:**
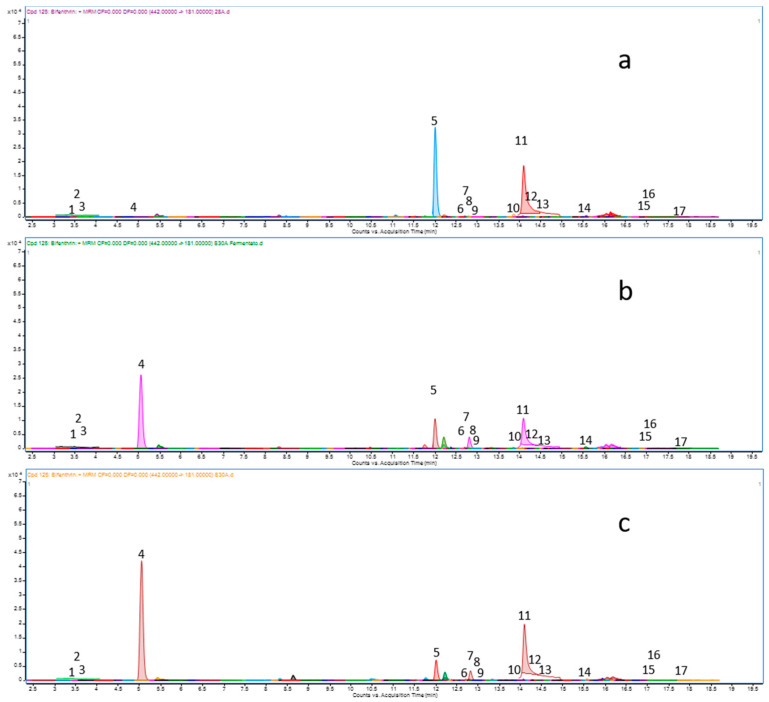
DMRM chromatograms of (**a**) grape juice, (**b**) must, (**c**) wine.

**Table 1 foods-10-01295-t001:** Linearities, curves, LODs, and LOQs and method validation parameters for the analysis of 115 target a.i. in grapes juice, must and wine in LC-MS/MS.

Pesticide	Linearity	Linear Regression Equation ^y^	r^2^ ± RSD% ^y^	MRL	LOD	LOQ	Apparent Recovery (%, *n* = 18) ^y^		RSD_r_ (3 × LOQ) ^y^	RSD_wR_ (3 × LOQ) ^y^	U *^,y^
	(μg kg^−1^)			(mg kg^−1^)	(μg kg^−1^)	(μg kg^−1^)	LOQ	5 × LOQ	*n* = 18	*n* = 36	
Cyromazine	LOQ—410	y = 9,931,209.1x + 52,051.8	0.9998 ± 0.17	0.05	1.37	4.10	107.1 ± 12.9	100.1 ± 11.7	10.9	9.9	30.0
Methamidophos	LOQ—410	y = 4,887,931x + 119,357.3	0.9999 ± 0.05	0.01	1.37	4.10	99.4 ± 8.3	95.6 ± 6.8	11.8	11.5	23.3
Acephate	LOQ—440	y = 375,881x − 24,719.0	0.9969 ± 0.17	0.01	1.45	4.40	93.1 ± 18.3	92.7 ± 5.4	12.7	10.6	14.7
Formetanate	LOQ—410	y = 14,913,520x − 37,835.2	0.9998 ± 0.12	0.1	1.38	4.10	78.5 ± 9.9	77.6 ± 6.3	14.5	13.2	20.6
Pymetrozine	LOQ—470	y = 696,973x − 11,370.1	0.9976 ± 0.16	0.02	1.58	4.70	109.4 ± 11.7	99.7 ± 7.8	13.2	13.3	15.1
Omethoate	LOQ—620	y = 32,351x + 372	0.9997 ± 0.08	0.01	2.07	6.20	81.4 ± 6.5	86.1 ± 3.9	13.9	13.9	38.2
Propamocarb	LOQ—420	y = 43,247,201x − 8829	1.0000 ± 0.21	0.01	1.39	4.10	96.9 ± 9.7	94.1 ± 8.7	11.3	9.4	30.3
Oxamyl	LOQ—400	y = 5,159,083x − 5579	0.9999 ± 0.09	0.01	1.34	4.00	93.6 ± 4.1	82.9 ± 6.3	6.7	6.0	37.9
Methomyl	LOQ—410	y = 5,917,725x + 7746	1.0000 ± 0.10	0.01	1.37	4.10	109.1 ± 7.4	105.7 ± 10.3	10.4	9.2	31.0
Flonicamid	LOQ—500	y = 784,761x + 1544	0.9982 ± 0.10	0.03	1.31	5.00	90.4 ± 11.6	92.4 ± 5.9	13.4	14.4	13.7
Thiamethoxam	LOQ—520	y = 11,189,678x + 71,281	0.9972 ± 0.41	0.40	1.75	5.20	101.8 ± 11.3	100.9 ± 7.5	9.9	9.7	34.6
Carbendazim	LOQ—400	y = 5,329,930x + 3082	0.9996 ± 0.16	0.30	1.33	4.00	75.8 ± 9.7	80.1 ± 3.1	8.6	8.0	15.6
Monocrotophos	LOQ—410	y = 17,644,763x + 44,731	0.9984 ± 0.19	0.01	1.38	4.10	93.4 ± 4.1	85.6 ± 8.4	8.6	7.5	40.4
Chlordimeform	LOQ—400	y = 853,728x − 1979	1.0000 ± 0.14	-	1.32	4.00	81.9 ± 8.4	76.9 ± 10.7	12.8	13.5	13.7
Cypermethrin	LOQ—410	y = 5,132,597x + 20,901	0.9964 ± 0.12	0.50 *	1.73	4.10	87.5 ± 10.1	85.4 ± 6.4	12.2	12.4	36.6
Imidacloprid	LOQ—320	y = 5,358,564x + 20,834	0.9999 ± 0.02	1.00 *	1.07	3.20	100.9 ± 1.7	99.5 ± 5.2	5.4	4.8	12.6
Methiocarb	LOQ—390	y = 31,899,495x + 73,553	0.9977 ± 0.19	0.30	1.29	3.90	75.6 ± 9.9	81.4 ± 9.2	7.3	6.4	31.0
Dimethoate	LOQ—410	y = 9,144,917x + 45,650	0.9999 ± 0.21	0.01	1.36	4.10	103.5 ± 10.0	94.8 ± 4.7	8.4	7.2	33.4
Acetamiprid	LOQ—400	y = 13,742,437x + 39,503	0.9998 ± 0.43	0.50 *	1.33	4.00	94.2 ± 3.7	89.7 ± 3.8	10.3	11.3	24.3
Cymoxanil	LOQ—440	y = 7,081,921x + 5097	0.9973 ± 0.34	0.30 *	1.45	4.40	97.5 ± 6.8	83.5 ± 7.8	13.1	12.2	35.2
Thiacloprid	LOQ—440	y = 4,714,164x + 21,318	0.9997 ± 0.26	0.01	1.46	4.40	104.7 ± 9.7	100.9 ± 9.2	12	11.6	29.1
Atrazine-desethyl	LOQ—430	y = 8,730,484x + 17,192	0.9994 ± 0.18	-	1.43	4.30	100.9 ± 6.2	98.7 ± 5.8	9.9	9.1	12.3
Aldicarb	LOQ—460	y = 22,774x − 3	0.9996 ± 0.42	0.02	1.53	4.60	97.5 ± 10.2	91.4 ± 6.2	9.1	9.5	32.4
Pirimicarb	LOQ—410	y = 30,848,679x + 25,212	0.9997 ± 0.16	0.01 *	1.37	4.10	103.1 ± 7.6	100.9 ± 11.8	8.6	9.0	15.6
Dichlorvos	LOQ—410	y = 450,147x + 175	1.0000 ± 0.17	0.01	1.37	4.10	81.9 ± 8.3	79.5 ± 6.0	12.9	10.8	17.5
Thiophanate-methyl	LOQ—410	y = 17,268,518x − 53,036	0.9993 ± 0.30	3.00 *	1.35	4.10	94.6 ± 15.0	91.7 ± 7.6	13.0	15.5	21.2
Metribuzin	LOQ—430	y = 3,761,325x + 17,279	0.9998 ± 0.47	0.10 *	1.42	4.30	93.7 ± 17.7	103.7 ± 4.4	9.0	7.3	20.5
Carbofuran	LOQ—420	y = 27,252,840x + 33,458	0.9990 ± 0.56	0.002	1.39	4.20	91.8 ± 9.9	90.9 ± 6.5	8.1	8.3	21.2
Carbaryl	LOQ—410	y = 14,046,644x + 7098	0.9972 ± 0.28	0.01	1.37	4.10	115.9 ± 8.0	99.7 ± 9.1	9.9	9.1	47.6
Imazalil	LOQ—430	y = 972,258x + 947	0.9995 ± 0.19	0.01 *	1.44	4.30	75.8 ± 3.2	75.9 ± 4.2	14.8	13.9	29.6
Fosthiazate	LOQ—400	y = 720,338x + 1656	0.9997 ± 0.05	0.02 *	1.32	4.00	94.1 ± 6.1	91.8 ± 8.1	9.1	8.2	20.8
Disulfoton-Sulfoxide	LOQ—470	y = 12,702,349x + 19,964	0.9999 ± 0.65	-	1.57	4.70	107.4 ± 9.0	101.9 ± 6.2	10.5	7.5	19.9
Flutriafol	LOQ—500	y = 3625x + 183	0.9982 ± 0.01	1.50 *	1.57	5.00	99.4 ± 11.1	97.5 ± 4.0	13.8	14.5	39.0
Metalaxyl	LOQ—390	y = 4371x + 183	0.9991 ± 0.19	1.00 *	1.30	3.90	78.9 ± 8.9	75.9 ± 6.1	9.8	9.6	26.6
Methidathion	LOQ—420	y = 396,329x − 101	0.9970 ± 0.10	0.02	1.41	4.20	96.7 ± 9.4	90.7 ± 8.9	11.5	9.4	30.4
Pyrimethanil	LOQ—390	y = 1,208,059x − 4250	0.9998 ± 0.06	5.00 *	1.32	3.90	101.6 ± 7.5	99.8 ± 4.3	10.6	10.4	15.9
Azinphos-methyl	LOQ—460	y = 398,216x + 390	0.9998 ± 0.46	0.05	1.55	4.60	99.4 ± 4.9	91.7 ± 9.3	13.9	11.2	20.9
Chlorantraniliprole	LOQ—400	y = 585,240x + 2	0.9991 ± 0.32	1.00 *	1.33	4.00	87.5 ± 3.8	85.4 ± 7.5	11.5	9.4	19.5
Diethofencarb	LOQ—390	y = 17,914,008x +75,801	0.9993 ± 0.78	0.90	1.28	3.90	95.9 ± 10.2	91.8 ± 3.6	13.2	9.9	37.6
Azoxystrobin	LOQ—430	y = 37027632 + 20,546	0.9981 ± 0.39	3.00 *	1.44	4.30	100.8 ± 8.5	97.6 ± 9.2	10.9	8.6	39.0
Propanil	LOQ—490	y = 1,172,102x − 4083	0.9998 ± 0.15	0.01	1.31	4.90	91.7 ± 4.1	89.9 ± 7.8	12.7	12.6	28.0
Fenamidone	LOQ—390	y = 19,532,597x + 43,718	1.0000 ± 1.00	0.60	1.30	3.90	96.8 ± 3.5	94.7 ± 10.3	9.5	9.1	24.7
Diclobutrazol	LOQ—390	y = 145,538x + 476	0.9988 ± 0.27	-	1.30	3.90	99.8 ± 10.7	98.4 ± 5.2	11.9	10.6	34.1
Boscalid	LOQ—410	y = 4,252,887x + 877	0.9995 ± 0.46	5.00 *	1.36	4.10	104.5 ± 5.6	101.7 ± 9.3	14.8	14.4	24.4
Dimethomorph	LOQ—390	y = 14,639,124x + 39,105	0.9989 ± 0.51	3.00 *	1.31	3.90	93.8 ± 7.4	90.7 ± 7.5	10.2	7.9	24.1
Benthiavalicarb	LOQ—240	y = 4,160,577x + 5366	0.9998 ± 0.23	0.30 *	0.78	2.40	111.4 ± 6.8	104.8 ± 3.9	8.8	8.8	25.4
Mandipropamid	LOQ—460	y = 5,148,550x − 20,963	0.9998 ± 0.78	2.00 *	1.52	4.60	85.6 ± 1.9	80.9 ± 5.4	10.0	10.6	31.5
Molinate	LOQ—620	y = 47,567x − 140	0.9999 ± 0.58	0.01	2.07	6.20	92.2 ± 7.1	93.7 ± 9.6	14.7	12.7	19.2
Chloroxuron	LOQ—430	y = 12,252,052x − 45,885	0.9997 ± 0.91	0.01	1.42	4.30	105.7 ± 9.7	99.4 ± 7.4	10.6	10.1	28.5
Bifenazate	LOQ—400	y = 1,008,299x + 76,836	0.9983 ± 0.47	0.70 *	1.33	4.00	87.0 ± 4.1	85.4 ± 4.6	11.7	9.9	29.8
Triadimenol	LOQ—500	y = 3,828,694x + 27.482	0.9989 ± 1.67	0.30	1.65	5.00	84.9 ± 7.3	90.7 ± 9.1	7.5	6.0	22.0
Cyproconazole	LOQ—410	y = 14,106,016x + 110,236	0.9980 ± 0.15	0.02 *	1.37	4.10	109.2 ± 6.2	110.7 ± 7.1	12.4	9.4	40.3
Iprovalicarb	LOQ—410	y = 4,408,560x + 40,826	0.9999 ± 0.19	2.00 *	1.36	4.10	90.6 ± 10.0	94.5 ± 6.4	13.5	13.8	29.9
Fenhexamid	LOQ—440	y = 3,699,942x + 18,108	0.9975 ± 1.00	15.00 *	1.45	4.40	112.4 ± 9.9	100.7 ± 4.6	10.5	8.0	13.2
Azinphos-ethyl	LOQ—390	y = 367,433x + 492	0.9976 ± 0.49	0.02	1.30	3.90	79.8 ± 5.5	75.9 ± 9.5	12.0	10.2	14.4
Myclobutanil	LOQ—410	y = 123,049x − 122	0.9994 ± 0.24	1.50*	1.37	4.10	78.5 ± 9.1	86.1 ± 3.7	10.6	9.1	39.1
Tetraconazole	LOQ—520	y = 9,182,219x − 30,061	0.9994 ± 0.17	0.50 *	1.72	5.20	109.7 ± 6.4	99.8 ± 7.9	11.5	9.5	38.8
Bupirimate	LOQ—400	y = 7,147,770x + 5118	0.9992 ± 0.54	1.50 *	1.32	4.00	99.4 ± 14.5	94.3 ± 9.2	10.9	7.1	26.1
Spirotetramat	LOQ—500	y = 2,820,842x + 73,839	0.9980 ± 0.92	2.00 *	1.68	5.00	81.8 ± 3.6	78.5 ± 7.7	16.2	14.5	39.2
Flufenacet	LOQ—410	y = 10,522,988x + 50,386	0.9981 ± 0.67	0.05 *	1.38	4.10	89.7 ± 4.6	86.7 ± 6.4	13.7	12.0	28.1
Mepanipyrim	LOQ—410	y = 328,904x − 348	0.9975 ± 1.01	2.00 *	1.36	4.10	99.4 ± 10.1	95.4 ± 4.6	12.1	8.7	35.4
Ethoprop	LOQ—380	y = 5,056,608x + 20,031	0.9972 ± 2.08	-	1.26	3.80	92.6 ± 5.1	96.7 ± 3.2	12.6	11.5	31.3
Napropamide	LOQ—410	y = 6,724,980x + 62,380	0.9969 ± 0.41	0.01 *	1.38	4.10	104.8 ± 6.2	103.4 ± 9.1	12.9	12.6	24.2
Cyazofamid	LOQ—410	y = 54,070x + 229	0.9988 ± 0.30	2.00 *	1.35	4.10	103.9 ± 11.0	98.9 ± 7.2	9.2	7.4	16.3
Cyprodinil	LOQ—410	y = 2,997,671x + 13,504	0.9994 ± 0.12	3.00 *	1.36	4.10	98.7 ± 5.1	97.1 ± 8.9	9.6	9.2	19.4
Flusilazole	LOQ—460	y = 9,842,141x + 54,831	0.9992 ± 0.85	0.01	1.52	4.60	110.1 ± 4.2	100.7 ± 6.2	11.1	9.8	27.8
Fenamiphos	LOQ—550	y = 2,610,314x + 116,029	0.9987 ± 0.24	0.03 *	1.82	5.50	99.7 ± 9.1	96.4 ± 4.2	11.6	8.1	24.7
Aclonifen	LOQ—390	y = 96,057x + 89	0.9967 ± 0.39	0.01 *	1.31	3.90	97.1 ± 6.2	99.1 ± 6.7	13.4	12.3	4.9
Penconazole	LOQ—400	y = 6,090,424x + 15,097	0.9963 ± 0.76	0.50 *	1.33	4.00	101.7 ± 4.2	99.7 ± 3.1	10.3	8.0	31.9
Tebuconazole	LOQ—400	y = 19,544,180x + 77,610	0.9984 ± 1.05	1.00*	1.34	4.00	99.5 ± 6.1	89.4 ± 7.1	8.9	7.1	20.5
Iprodione	LOQ—460	y = 82,937x + 89	0.9966 ± 0.64	0.01	1.52	4.60	93.4 ± 2.0	97.4 ± 9.2	12.8	11.3	17.6
Benalaxyl	LOQ—480	y = 23,788,991x + 92,142	0.9958 ± 0.87	0.30 *	1.60	4.80	100.0 ± 1.9	99.2 ± 6.1	9.0	8.4	48.2
Zoxamide	LOQ—410	y = 5,772,043x + 12,727	0.9984 ± 0.21	5.00 *	1.37	4.10	98.7 ± 6.4	99.0 ± 2.8	12.4	12.4	34.4
Spinosyn A	LOQ—430	y = 1,125,007x − 3076	0.9986 ± 0.54	0.50 *	1.44	4.30	75.6 ± 8.1	79.4 ± 7.6	10.2	9.6	36.6
Pyraclostrobin	LOQ—420	y = 13,354,734x − 5243	0.9968 ± 1.09	2.00 *	1.39	4.20	90.8 ± 10.2	95.1 ± 9.9	15.3	15.2	32.9
Cyflufenamid	LOQ—410	y = 5,945,864x + 38,431	0.9989 ± 0.45	-	1.36	4.10	81.9 ± 1.6	91.1 ± 4.1	8.7	7.9	28.4
Clofentezin	LOQ—410	y = 1,910,622x −533	0.9990 ± 0.73	1.00 *	1.36	4.10	85.7 ± 9.4	101.7 ± 10.8	9.1	9.0	23.9
Bitertanol	LOQ—410	y = 6,031,192x + 40,609	0.9988 ± 0.29	0.01	1.35	4.10	101.9 ± 6.7	99.7 ± 6.9	15.0	15.2	49.3
Phosalone	LOQ—580	y = 6,702,437x + 16,186	0.9970 ± 0.13	0.01	1.93	5.80	98.7 ± 4.4	91.7 ± 9.1	14.8	15.6	38.1
Metrafenone	LOQ—470	y = 9,239,760x + 20,728	0.9975 ± 0.56	7.00 *	1.56	4.70	99.1 ± 8.9	89.1 ± 7.9	12.9	12.0	33.9
Difenconazole	LOQ—490	y = 20,779,327x + 20,116	0.9974 ± 0.89	3.00 *	1.62	4.90	96.7 ± 2.3	91.7 ± 3.8	10.7	8.2	33.5
Chlorpyrifos-methyl	LOQ—410	y = 65,820x − 225	0.9974 ± 0.34	0.01	1.38	4.10	94.8 ± 3.9	99.6 ± 8.3	17.9	17.6	26.7
Ametoctradin	LOQ—320	y = 8,626,247x + 1875	0.9962 ± 0.46	6.00 *	1.06	3.20	99.7 ± 9.1	97.8 ± 10.4	8.5	8.7	31.9
Spinosyn D	LOQ—430	y = 209,477x − 615	0.9971 ± 0.24	0.50 *	1.44	4.30	89.5 ± 4.6	85.4 ± 6.1	8.1	5.7	39.1
Indoxacarb	LOQ—450	y = 1,790,071x − 2915	0.9993 ± 0.76	2.00 *	1.50	4.50	97.4 ± 9.5	91.6 ± 9.1	7.0	6.3	40.1
Cycloate	LOQ—500	y = 598,248x − 1864	0.9973 ± 0.50	-	1.67	5.00	106.9 ± 2.7	100.1 ± 3.1	9.1	10.9	36.9
Hexaflumuron	LOQ—420	y = 633,012x − 654	0.9985 ± 0.94	-	1.40	4.20	78.5 ± 15.2	79.4 ± 9.3	9.7	9.8	37.9
Trifloxystrobin	LOQ—430	y = 17,415,017x + 17,609	0.9956 ± 0.34	3.00 *	1.44	4.30	91.9 ± 4.3	95.9 ± 10.0	10.2	9.6	35.3
Quizalofop-ethyl	LOQ—400	y = 3,145,820x − 5888	0.9955 ± 0.78	0.02 *	1.33	4.00	102.4 ± 3.7	99.4 ± 11.5	13.5	10.8	44.4
Metaflumizone	LOQ—410	y = 949,238x − 6681	0.9971 ± 0.60	0.05 *	1.37	4.10	95.6 ± 6.8	96.1 ± 4.6	8	7.8	48.3
Buprofezin	LOQ—470	y = 20,507,228x + 51,039	0.9999 ± 0.24	0.01 *	1.55	4.70	93.4 ± 9.0	97.2 ± 9.7	12.7	11.4	21.7
Tebufenpyrad	LOQ—400	y = 4,341,528x + 15,573	0.9987 ± 0.53	0.60 *	1.34	4.00	80.9 ± 7.8	78.9 ± 6.8	13.6	11.0	46.7
Emamectin Benzoate	LOQ—520	y = 3,632,831x − 14,712	0.9985 ± 0.25	0.05 *	1.74	5.20	96.4 ± 11.2	98.1 ± 7.1	10.1	7.8	42.8
Propaquizafop	LOQ—430	y = 2,547,342x − 6777	0.9963 ± 0.12	0.02 *	1.44	4.30	96.8 ± 3.8	100.6 ± 8.5	13.1	13.6	47.5
Lufenuron	LOQ—440	y = 568,999x + 552	0.9998 ± 1.02	0.01 *	1.46	4.40	90.1 ± 9.4	94.9 ± 6.6	8.3	9.8	30.7
Oxadiazon	LOQ—400	y = 726,169x − 4189	0.9973 ± 0.98	0.05	1.34	4.00	91.7 ± 7.8	90.7 ± 5.4	9.9	11.1	35.9
Allethrin	LOQ—650	y = 439,397x − 5650	0.9999 ± 0.01	-	2.16	6.50	88.4 ± 10.3	79.1 ± 6.1	11.2	9.8	15.5
Piperonyl butoxide	LOQ—400	y = 31,536,094x + 11,745	0.9988 ± 0.28	-	1.35	4.00	105.7 ± 9.5	99.8 ± 4.8	11.8	11.2	32.0
Pyriproxyfen	LOQ—420	y = 6,633,898x − 13,776	0.9981 ± 0.85	0.05 *	1.39	4.20	80.9± 7.8	85.4 ± 9.5	9.5	9.6	25.4
Cycloxydim	LOQ—410	y = 137,254x + 708	0.9987 ± 0.57	0.50 *	1.36	4.10	87.1 ± 3.8	87.4 ± 7.8	15.7	15.8	40.2
Chlorpyriphos	LOQ—400	y = 638,457x − 2533	0.9981 ± 0.64	0.01	1.32	4.00	101.9 ± 8.1	96.7 ± 3.0	10.2	10.8	24.7
Hexythiazox	LOQ—360	y = 10,457,257x − 49,960	0.9985 ± 0.95	1.00 *	1.19	3.60	95.7 ± 9.8	100.5 ± 5.5	9.9	8.7	42.4
Pendimethalin	LOQ—330	y = 1,172,935x − 2381	0.9978 ± 0.46	0.05 *	1.11	3.30	87.6 ± 10.3	89.1 ± 8.6	8.4	8.4	20.6
Flufenoxuron	LOQ—390	y = 3,953,910x + 11,436	0.9984 ± 0.30	2.00	1.30	3.90	83.7 ± 3.4	79.7 ± 9.5	7.6	6.6	29.1
Propargite	LOQ—380	y = 6,439,838x + 13,171	0.9976 ± 0.76	0.01	1.27	3.80	81.8 ± 5.0	79.1 ± 4.2	14.3	15.5	27.8
Fenpyroximate(E)	LOQ—460	y = 24,740,316x + 67,928	0.9979 ± 0.12	0.30 *	1.53	4.60	99.8 ± 4.0	98.7 ± 6.2	9.4	10.4	27.3
Deltamethrin	LOQ—340	y = 212,931x + 1529	0.9976 ± 0.54	0.05	1.15	3.40	94.5 ± 9.7	93.4 ± 8.1	14.2	14.2	21.0
Acrinathrin	LOQ—470	y = 48,200x − 207	0.9972 ± 0.56	0.10 *	1.58	4.70	79.4 ± 10.2	80.9 ± 3.7	11.3	12.7	31.2
Etoxazole	LOQ—520	y = 393,488x + 312	0.9969 ± 0.75	0.50 *	1.72	5.20	90.7 ± 9.6	95.7 ± 7.7	9.6	9.6	36.5
Pyridaben	LOQ—420	y = 18,219,756x + 7898	0.9966 ± 0.32	0.01 *	1.39	4.20	99.8 ± 5.6	94.7 ± 8.2	14.3	14.2	22.5
Tau—Fluvalinate	LOQ—430	y = 17,740,157x − 22,927	0.9993 ± 0.46	1.00 *	1.44	4.30	100.8 ± 1.4	102.5 ± 6.8	17.7	18.0	22.9
Fenarimol	LOQ—440	y = 167,645x + 140	0.9971 ± 0.92	0.30	1.48	4.40	83.9 ± 6.4	89.8 ± 7.2	9.8	10.0	45.3
Etofenprox	LOQ—390	y = 5,530,818x + 37,481	0.9993 ± 0.15	4.00 *	1.29	3.90	97.1 ± 8.6	95.4 ± 9.5	7.5	6.1	17.5
Bifenthrin	LOQ—420	y = 39,808x + 128	0.9977 ± 1.46	0.30	1.39	4.20	89.1 ± 4.6	86.1 ± 11.4	6.8	6.1	20.0
Famoxadone	LOQ—380	y = 168,731x − 107	0.9984 ± 0.12	2.00 *	1.25	3.80	89.5 ± 8.9	82.8 ± 4.1	11.0	9.8	49.1

* Pesticides allowed in EU with an MRL on wine grapes. ^y^ validation data (linear regression equation, r^2^, Apparent recovery, RSD_r_ and RSD_wr_, and U) represent the average values from grape juice, must and wine.

**Table 2 foods-10-01295-t002:** Pesticide residues concentration in grape juice, must, and wine from 1–23 samples.

	Pesticide	N°	Vat 30 (mg/L—mg/kg)							
	Frequency	Samples	Grapes		Must		Wine		PF1	PF2
Pesticides	grapes (1–23)	>MRL	mg L^−1^	mg kg^−1^	mg L^−1^	mg kg^−1^	mg L^−1^	mg kg^−1^		
			Juice	Pellet	Liquid	Pellet	Liquid	Pellet		
Acephate	13		<LOQ	<LOQ	<LOD	<LOQ	<LOD	<LOQ		
Formetanate	13		<LOQ	<LOD	<LOQ	<LOD	<LOD	<LOQ		
Pymetrozine	13		<LOQ	<LOQ	<LOQ	<LOQ	<LOD	<LOQ		
Carbendazim	13		<LOQ	<LOD	<LOQ	<LOD	<LOD	<LOQ		
Mandipropoamid	23	3	<LOQ−6.60	<LOQ	1.05 ± 0.61	1.76 ± 12.21	1.55 ± 2.32	7.48 ± 4.05	0.23	1.48
Iprovalicarb	4		<LOQ	<LOD	<LOQ	<LOQ	<LOD	<LOQ		
Fenhexamid	22		<LOQ−0.03	<LOQ	<LOQ	<LOQ	<LOQ	0.617		
Azinphos-ethyl	16		<LOQ	<LOD	<LOQ	<LOQ	<LOD	<LOQ		
Mepanipyrim	16	1	<LOQ−2.36	<LOQ	<LOQ	<LOQ	<LOD	<LOQ		
Iprodione	8		<LOQ−0.14		<LOQ	<LOQ	<LOQ	<LOQ		
Zoxamide	13		<LOQ−0.15	2.02 ± 2.19	0.02 ± 2.06	0.09 ± 2.68	0.02 ± 2.06	1.09 ± 0.19	0.13	1.00
Spinosyn A	5		<LOQ–0.02	0.12 ± 1.48	<LOQ	0.09 ± 0.48	<LOD	0.90 ± 0.09	<0.04	nt
Spinosyn D	5		<LOQ	0.14 ± 1.23	<LOQ	0.04 ± 4.05	<LOQ	0.69 ± 0.17		
Emamectin Benzoate	10		<LOD	<LOQ	<LOD	0.13 ± 0.02	<LOD	2.13 ± 0.02		
Etoxazole	0		-	-	-	-	-	-		
T-fluvalinate	10		<LOQ	0.09 ± 3.87	<LOQ	0.04 ± 1.63	<LOQ	1.23 ± 4.04		
Etofenprox	1		0.08 ± 2.12	1.13 ± 0.64	<LOQ	<LOQ	<LOQ	<LOQ	<0.04	nt

PF1: processing factor grapes vs. wine; PF2: processing factor must vs. wine; nt: not derived.

**Table 3 foods-10-01295-t003:** Pesticide residues concentration in grape juice, must, and wine from 24–35 samples.

	Pesticide	N°	Vat 30 (mg/L—mg/kg)							
	Frequency	Samples	Grapes		Must		Wine		PF1	PF2
Pesticides	Grapes (24–35)	>MRL	mg L^−1^	mg kg^−1^	mg L^−1^	mg kg^−1^	mg L^−1^	mg kg^−1^		
			Juice	Pellet	Liquid	Pellet	Liquid	Pellet		
Acephate	3		<LOQ	<LOQ	<LOD	<LOQ	<LOD	<LOQ		
Formetanate	3		<LOQ	<LOQ	<LOQ	<LOQ	<LOD	<LOQ		
Pymetrozine	3		<LOQ	<LOQ	<LOQ	<LOQ	<LOD	<LOQ		
Carbendazim	3		<LOQ	<LOQ	<LOQ	<LOQ	<LOD	<LOQ		
Mandipropoamid	6	3	<LOQ-10.56	<LOQ	0.75 ± 3.99	1.59 ± 2.32	1.01 ± 0.43	4.19 ± 2.32	0.10	1.35
Iprovalicarb	0		-	-	-	-	-	-		
Fenhexamid	1		0.35 ± 2.34	<LOQ	<LOQ	<LOQ	<LOD	0.56 ± 0.21	<0.04	nd
Azinphos-ethyl	3		<LOQ	<LOQ	<LOQ	<LOQ	<LOD	<LOQ		
Mepanipyrim	3		<LOQ-0.60	<LOQ	<LOQ	<LOQ	<LOD	<LOQ	<0.04	nd
Iprodione	2		<LOQ-0.03	0.11 ± 0.26	<LOQ	<LOQ	<LOQ	<LOQ	<0.05	nd
Zoxamide	3		<LOQ-0.08	1.26 ± 4.70	<LOQ-0.02	0.08 ± 4.52	0.02 ± 6.71	1.18 ± 0.45	0.25	1.00
Spinosyn A	5		<LOQ–0.03	0.13 ± 0.81	<LOQ	0.05 ± 5.51	<LOD	0.89 ± 5.51	<0.04	nd
Spinosyn D	5		<LOQ	0.07 ± 0.11	<LOQ	0.04 ± 4.84	<LOQ	0.71 ± 1.12		
Emamectin Benzoate	5		<LOQ-0.04	0.31 ± 0.03	<LOQ-0.02	0.11 ± 11.26	0.021	2.10 ± 0.03	0.53	1.05
Etoxazole	2		<LOQ-0.03	<LOQ	<LOQ	<LOQ	<LOQ	<LOQ	<0.05	nd
T-fluvalinate	5		<LOQ	0.12 ± 3.16	<LOQ	0.04 ± 4.41	<LOQ	0.90 ± 4.04		
Etofenprox	0		-	-	-	-	-	-		

PF1: processing factor grapes vs. wine; PF2: processing factor must vs. wine; nd: not derived.

## Data Availability

Not applicable.

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
