# Peer review of "Effect of the Technological Process from Vine to Wine on Pesticide Residues in Vernaccia di Oristano Cultivar"

_foods, 2021, doi:10.3390/foods10061295_

Round 1

Reviewer 1 Report

  1. 29 we cannot speak about healthy levels for pesticide residues. Change text for instance:

… winemaking to decrease pesticide residues in wine compared to the raw material

  1. 57 multi residue analyses will not ensure food safety (will they distroy the valuable grape if the residues are detected?

L.83 I assume that the grape samples were collected before crushing

L.157 change to: complying with SANTE principle

  1. 167. Pls use metric units insead of quintals

It is suggested that the authors determine the processing factors from grape tp wine and from must to wine according tot he principles used by the FAO/WHO JMPR and recommended by the OECD Working Group on Pesticide Residue . It would make their findings clearer and comparable to other grape processing results.

Pls see the highlighted parts

Author Response

Reviewer 1

R: 29 we cannot speak about healthy levels for pesticide residues. Change text for instance: … winemaking to decrease pesticide residues in wine compared to the raw material.

Answer: the text was corrected as requested (Line 26)

R: 57 multi residue analyses will not ensure food safety (will they destroy the valuable grape if the residues are detected?)

Answer: the text was improved as requested (Line 53-54)

R: 83 I assume that the grape samples were collected before crushing

Answer: the text was improved (Line 72 and line 81-83)

R: 157 change to: complying with SANTE principle

Answer: the text was modified as requested (Line 156)

R: 167. Pls use metric units instead of quintals.

Answer: the text was modified as requested (Line 166)

R: It is suggested that the authors determine the processing factors from grape to wine and from must to wine according to the principles used by the FAO/WHO JMPR and recommended by the OECD Working Group on Pesticide Residue. It would make their findings clearer and comparable to other grape processing results.

Answer: the text was improved with PF information (table 2, 3, and line 242-246)

R: Pls see the highlighted parts

Answer: see previous answers and line 263-265.

Reviewer 2 Report

The study focused on the determination of residues of 115 pesticide in thirty-five samples of grapes, two musts, and two vines by a multiresidue QuEChERS method and UPLC-MS/MS to assess the effect of the technological process from vine to wine. This research is not particularly original (e.g., QuEChERS method and the UPLC-MS/MS method is the same that the authors previously used to assess the effect of the process of raw tomatoes (Foods 2020, 9, 1497; doi:10.3390/foods9101497)). Moreover, the manuscript shows other lacks:

  • Subsection 2.1. Sampling process should be described.
  • Data about fermentation vat V31 were missing. This should be provided at least as supplementary material.
  • Section 2.3 and 2.4 should be merged and summarized, indicating only the parameter that differ from previous work (Foods 2020, 9, 1497; doi:10.3390/foods9101497) (i.e., 5 mL of samples). Moreover, reference 12 should properly mentioned.
  • Table 1. Not allowed pesticides should be indicated.
  • Table 1. Apparent recovery? This should be clarified in the manuscript.
  • Line 55. The reference https://doi.org/10.1016/B978-0-444-63430-6.00015-1 should be included.
  • Line 220. References should be provided.
  • References section. References 27-31 are missing.

Author Response

Reviewer 2

R: Subsection 2.1. Sampling process should be described.

Answer: the text was improved (Line 72 and line 81-83)

R: Data about fermentation vat V31 were missing. This should be provided at least as supplementary material.

Answer: I did not understand what the referee means.

R: Section 2.3 and 2.4 should be merged and summarized, indicating only the parameter that differ from previous work (Foods 2020, 9, 1497; doi:10.3390/foods9101497) (i.e., 5 mL of samples). Moreover, reference 12 should properly mentioned.

Answer: the text was modified accordingly (line 106-111)

R: Table 1. Not allowed pesticides should be indicated.

Answer: the table indicates the pesticides allowed; all other pesticides are not allowed.

R: Table 1. Apparent recovery? This should be clarified in the manuscript.

Answer: the use of the term apparent recovery is recommended when the observed value is obtained via an analytical procedure such as a calibration graph (DUNCAN THORBURN BURNS, KLAUS DANZER, AND ALAN TOWNSHEND. IUPAC Recommendations 2002)

R: Line 55. The reference https://doi.org/10.1016/B978-0-444-63430-6.00015-1 should be included.

Answer: the reference was included in the text and in reference section.

R: Line 220. References should be provided.

Answer: the reference was added

R: References section. References 27-31 are missing.

Answer: the references were added, and the correct number of each reference was checked in the text.

Round 2

Reviewer 2 Report

In my judgment, after reviewing carefully the manuscript, the confusing issues have been clarified. Currently, the article includes all the necessary information for a proper understanding of the work.

Only minor change needs to be made: The meaning of “apparent recovery” should also be included in the manuscript together with the reference about this issue ((IUPAC Recommendations 2002 by Duncan Thorburn Burns , Klaus Danzer , Alan Townshend).

Author Response

The meaning of “apparent recovery” was included in the manuscript, and the reference was added to the reference list.